# AutoGluon–TimeSeries:
# AutoML for Probabilistic Time Series Forecasting

Oleksandr Shchur[1]  Caner Turkmen[1]  Nick Erickson[1]  Huibin Shen[2]  Alexander Shirkov[1]
Tony Hu[1]  Yuyang Wang[2]

[1]Amazon Web Services
[2]AWS AI Labs

**Abstract**  We introduce AutoGluon–TimeSeries—an open-source AutoML library for probabilistic time series forecasting.[1] Focused on ease of use and robustness, AutoGluon–TimeSeries enables users to generate accurate point and quantile forecasts with just 3 lines of Python code. Built on the design philosophy of AutoGluon, AutoGluon–TimeSeries leverages ensembles of diverse forecasting models to deliver high accuracy within a short training time. AutoGluon–TimeSeries combines both conventional statistical models, machine-learning based forecasting approaches, and ensembling techniques. In our evaluation on 29 benchmark datasets, AutoGluon–TimeSeries demonstrates strong empirical performance, outperforming a range of forecasting methods in terms of both point and quantile forecast accuracy, and often even improving upon the best-in-hindsight combination of prior methods.

## 1 Introduction

Time series (TS) forecasting is a fundamental statistical problem with applications in diverse domains such as inventory planning (Syntetos et al., 2009), smart grids (Hong et al., 2020), and epidemiology (Nikolopoulos et al., 2021). Decades of research led to development of various forecasting approaches, from simple statistical models (Hyndman and Athanasopoulos, 2018) to expressive deep-learning-based architectures (Benidis et al., 2022). Despite the availability of various forecasting approaches, practitioners often struggle with selecting the most appropriate method and adhering to best practices when implementing and evaluating forecasting pipelines.

AutoML aims to mitigate these challenges by providing tools that enable practitioners to develop accurate and efficient predictive models without extensive domain knowledge. While traditional AutoML methods have focused primarily on classification and regression tasks for tabular data (Thornton et al., 2013; Feurer et al., 2015; Olson and Moore, 2016; Erickson et al., 2020; LeDell and Poirier, 2020; Zimmer et al., 2021), automated time series forecasting has received comparatively less attention, with only a few open-source AutoML forecasting frameworks having been proposed (Deng et al., 2022; Catlin, 2022). Furthermore, existing automated forecasting frameworks tend to generate point forecasts without considering uncertainty, which is a crucial factor in many practical applications (Gneiting and Katzfuss, 2014).

To close this gap, we introduce AutoGluon–TimeSeries (AG–TS), an open-source AutoML framework for probabilistic time series forecasting written in Python. AG–TS can generate both point and probabilistic forecasts for collections of univariate time series. Together with support for static and time-varying covariates, this makes AG–TS applicable to most real-world forecasting tasks.

As part of the AutoGluon framework (Erickson et al., 2020; Shi et al., 2021), AG–TS adheres to the principles of ease of use and robustness, empowering users with limited expertise in the target domain to generate highly accurate predictions with minimal coding effort. The architecture is

---

[1]https://github.com/autogluon/autogluon

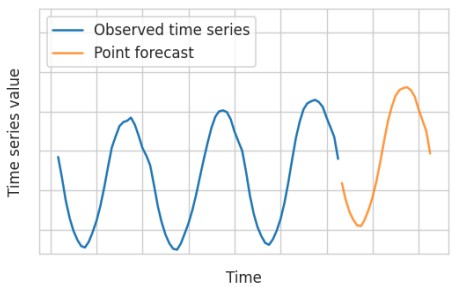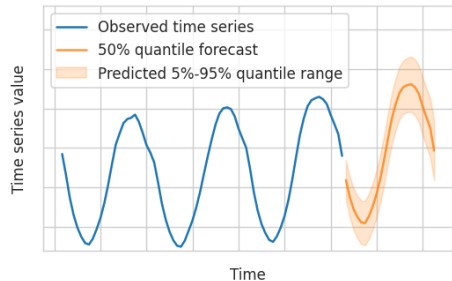

Figure 1: Point forecast (left) and quantile forecast (right) for a univariate time series.

capable of handling failures of individual models when necessary, producing a valid result as long as any single model was trained successfully.

We evaluate the performance of AG–TS against other established forecasting methods and AutoML systems using 29 publicly available benchmark datasets. The results demonstrate AG–TS's strong performance, outperforming various competing approaches in terms of both point and probabilistic forecast accuracy. This highlights the potential of AG–TS as a valuable tool for practitioners and researchers seeking an automated and versatile solution for time series forecasting.

## 2 Probabilistic Time Series Forecasting

The probabilistic time series forecasting problem can be formally stated as follows. The data $\mathcal{D} = \{\boldsymbol{y}_{i,1:T_i}\}_{i=1}^{N}$ is a collection of $N$ univariate time series, where $\boldsymbol{y}_{i,1:T_i} = (y_{i,1}, ..., y_{i,T_i})$, $y_{i,t}$ is the value of the $i$-th time series at time $t$, and $T_i$ is the length of the $i$-th time series.[2] For example, $y_{i,t}$ may correspond to the number of units of product $i$ sold on day $t$. The goal of time series forecasting is to predict the future $H$ values for each time series in $\mathcal{D}$. The parameter $H$ is known as *prediction length* or *forecast horizon*.

Each time series $\boldsymbol{y}_{i,1:T}$ may additionally be associated with covariates $\boldsymbol{X}_{i,1:T+H}$. These include both *static covariates* (e.g., location of the store, product ID) and *time-varying covariates*. The time-varying covariates may, in turn, be known in the future (e.g., day of the week, promotions) or only known in the past (e.g., weather, sales of other products).

In the most general form, the goal of probabilistic forecasting is to model the conditional distribution of the future time series values $\boldsymbol{y}_{i,T+1:T+H}$ given the past values $\boldsymbol{y}_{i,1:T}$ and the related covariates $\boldsymbol{X}_{i,1:T+H}$

$$p(\boldsymbol{y}_{i,T+1:T+H} \mid \boldsymbol{y}_{i,1:T}, \boldsymbol{X}_{i,1:T+H}).$$

In practice, we are rarely interested in the full predictive distribution and rather represent the range of possible outcomes with *quantile forecasts* $\hat{\boldsymbol{y}}_{i,T+1:T+H}^{q}$ for chosen quantile levels $q \in (0, 1)$. The quantile forecast implies that the future time series value $y_{i,T+h}$ is predicted to exceed $\hat{y}_{i,T+h}^{q}$ with probability $q$ (Wen et al., 2017; Lim et al., 2021).

If the uncertainty is of no interest, we can instead report a *point forecast* of the future time series values. For example, we can summarize the prediction using the conditional mean

$$\hat{\boldsymbol{y}}_{i,T+1:T+H} = \mathbb{E}_p[\boldsymbol{y}_{i,T+1:T+H} \mid \boldsymbol{y}_{i,1:T}, \boldsymbol{X}_{i,1:T+H}].$$

Figure 1 demonstrates the difference between a point forecast and a quantile forecast. Finally, note that here we consider the problem of forecasting multiple univariate time series, also known as panel data, which is different from multivariate forecasting (Benidis et al., 2022).

---

[2]To reduce clutter in notation, we assume that all time series have the same length $T$ (even though AG–TS supports the case when time series have different lengths).

## 3 AutoGluon–TimeSeries

AutoGluon–TimeSeries enables users to generate probabilistic time series forecasts in a few lines of code, as shown by the following minimal example.

```python
from autogluon.timeseries import TimeSeriesDataFrame, TimeSeriesPredictor

train_data  = TimeSeriesDataFrame.from_path("train.csv")
predictor   = TimeSeriesPredictor(prediction_length=30).fit(train_data)
predictions = predictor.predict(train_data)  # forecast next 30 time steps
```

**Loading the data**. A TimeSeriesDataFrame object stores a collection of univariate time series and provides utilities such as loading data from disk and train-test splitting. Internally, time series data is represented as a pandas.DataFrame (pandas development team, 2020) in long format (Table 1), but loaders are also available for other formats. Besides the target time series that need to be forecast, TimeSeriesDataFrame can also store the static and time-varying covariates.

Table 1: Collection of univariate time series stored as a TimeSeriesDataFrame. Each row contains unique ID of the time series, timestamp, and the value of the target time series.

| item_id | timestamp | target |
|---:|---:|---:|
| T1 | 2020-03-02 | 23 |
| T1 | 2020-03-03 | 43 |
| ... | ... | ... |
| T999 | 2020-08-29 | 15 |
| T999 | 2020-08-31 | 27 |

**Defining the task**. Users can specify the forecasting task by creating a TimeSeriesPredictor object. Task definition includes information such as *prediction length*, list of *quantile levels* to be predicted, and the *evaluation metric*. The evaluation metric should be chosen based on the downstream application. For example, mean weighted quantile loss (wQL) measures the accuracy of *quantile* forecasts, and mean absolute scaled error (MASE) reports the accuracy of the *point* forecast relative to a naive baseline. When creating the predictor, users can also specify what time-varying covariates are known in the future—the remainder will be treated as past-only covariates.

**Fitting the predictor**. Inside the fit() method, the predictor preprocesses the data, fits and evaluates various models using cross-validation, optionally performs hyperparameter optimization (HPO) on selected models, and trains an ensemble of the individual forecasting models. By default, AG–TS provides user-friendly *presets* users can choose from to manage the training time–accuracy tradeoff. Advanced users can also explicitly specify the models to use and their hyperparameters, or specify search spaces in which optimal hyperparameters will be searched.

**Making predictions**. After the predictor has been fit, the predict() method can be used to generate predictions on new data—including time series that haven't been seen during training. Like the input data, the predictions are stored in a long-format data frame, where the columns contain the mean (expected value) and quantile forecasts at the desired quantile levels (Table 2).

**Documentation**. We provide various additional resources on the official website auto.gluon.ai. These include installation instructions, tutorials, and a cheatsheet summarizing the main features.

### 3.1 Design Considerations

AG–TS was launched as a part of the AutoGluon suite (Erickson et al., 2020) in v0.5, building on the foundation of AutoGluon and borrowing some design elements from other forecasting libraries like GluonTS (Alexandrov et al., 2020). Since then, AG–TS has evolved into a full solution for time series forecasting. Below, we highlight some of AG–TS's key design principles.

Table 2: Mean and quantile forecasts generated by a `TimeSeriesPredictor`. The forecasts include the next `prediction_length` many time steps of each time series in the dataset.

| item_id | timestamp | mean | 0.1 | 0.5 | 0.9 |
|--------:|-----------|-----:|----:|----:|----:|
| T1 | 2020-09-01 | 17 | 10 | 16 | 23 |
| T1 | 2020-09-02 | 25 | 15 | 23 | 31 |
| ... | ... | ... | ... | ... | ... |
| T999 | 2020-09-29 | 33 | 21 | 33 | 36 |
| T999 | 2020-09-30 | 30 | 24 | 28 | 34 |

**Ensembles over HPO**. AG–TS follows the AutoGluon philosophy, relying on ensembling techniques instead of HPO or neural architecture search. The library features a broad selection of models whose probabilistic forecasts are combined in an ensemble selection step (Caruana et al., 2004). AG–TS favors broadening the portfolio of forecasters over exploring the hyperparameter space of any particular model. While AG–TS does support HPO techniques, HPO is excluded from most preset configurations to reduce training time and minimize overfitting on the validation data.

**Presets and default hyperparameters**. In order to provide defaults that work well out of the box for users that are not familiar with forecasting, AG–TS includes various *presets*—high-level configuration options that allow users to trade off between fast training and higher accuracy. AG–TS follows the convention-over-configuration principle: all models feature default configurations of hyperparameters that are expected to work well given the selected preset. At the same time, advanced users have an option to manually configure individual models and use the `TimeSeriesPredictor` as a unified API for training, evaluating and combining various forecasting models (see documentation for details).

**Model selection**. Time series forecasting introduces unique challenges in model validation and selection. Importantly, as the main aim of the model is to generalize *into the future*, special care has to be taken to define validation sets that are held out *across time*. The AG–TS API is designed with this consideration. If the user does not explicitly specify a validation set, the library holds the window with last `prediction_length` time steps of each time series as a validation set. Optionally, multiple windows can be used to perform so-called *backtesting*.

## 3.2 Forecasting Models

There are two families of approaches to forecasting in large panels of time series. The first approach is to fit *local* classical parametric statistical models to each individual time series. A second approach is built on expressive machine-learning-based approaches that are fit *globally* on all time series at once. AG–TS features both approaches, incorporating forecasting models from both families and combining them in an ensemble.

**Local models**. This category contains conventional methods that capture simple patterns like trend and seasonality. Examples include *ARIMA* (Box et al., 1970), *Theta* (Assimakopoulos and Nikolopoulos, 2000) and *ETS* (Hyndman et al., 2008), as well as simple baselines like *Seasonal Naive* (Hyndman and Athanasopoulos, 2018). AG–TS relies on implementations of these provided by `StatsForecast` (Garza et al., 2022).

The defining characteristic of local models is that a separate model is fit to each individual time series in the dataset (Januschowski et al., 2020). This means that local models need to be re-fit when making predictions for new time series not seen during training. To mitigate this limitation, AG–TS caches the model predictions and parallelizes their fitting across CPU cores using Joblib (Joblib Development Team, 2020).

**Global models.** Unlike local models, a single global model is fitted to the entire dataset and used to make predictions for all time series. Global models used by AG–TS can be subdivided into two categories: deep learning and tabular models. Deep-learning models such as *DeepAR* (Salinas et al., 2020), *PatchTST* (Nie et al., 2023), and *Temporal Fusion Transformer* (Lim et al., 2021) use neural networks to generate probabilistic forecasts for future data. AG–TS uses PyTorch-based deep learning models from `GluonTS` (Alexandrov et al., 2020). Tabular models like *LightGBM* (Ke et al., 2017) operate by first converting the time series forecasting task into a tabular regression problem. This can be done either *recursively*—by predicting future time series values one at a time—or by *directly* forecasting all future values simultaneously (Januschowski et al., 2022). AG–TS relies on regression models provided by AutoGluon–Tabular and uses `MLForecast` (Nixtla, 2023) for converting them into tabular forecasters.

Global models typically provide faster inference compared to local models, since there is no need for re-training at prediction time. This, however, comes at the cost of longer training times since more parameters need to be estimated. Global models also naturally handle various types of covariates and utilize information present across different time series, which is known as cross-learning (Semenoglou et al., 2021).

**Ensembling.** After AG–TS finishes sequentially fitting the individual models, they are combined using 100 steps of the forward selection algorithm (Caruana et al., 2004). The output of the ensemble is a convex combination of the model predictions:

$$\hat{y}_{i,T+1:T+H}^{\text{ensemble}} = \sum_{m=1}^{M} w_m \cdot \hat{y}_{i,T+1:T+H}^{(m)} \qquad \text{subject to } w_m \geq 0, \sum_{m=1}^{M} w_m = 1,$$

where $\hat{y}_{i,T+1:T+H}^{(m)}$ are either point or quantile forecasts generated by each of the $M$ trained models. Note that in case of probabilistic forecasting, the ensemble computes a weighted average of the quantile forecasts of the individual models—method known as Vincentization (Ratcliff, 1979).

The ensemble weights $w_m$ are tuned to optimize the chosen evaluation metric (e.g., wQL, MASE) on the out-of-fold predictions generated using time series cross-validation (Hyndman and Athanasopoulos, 2018). The main advantages of the forward selection algorithm are its simplicity, compatibility with arbitrary evaluation metrics, and the sparsity of the final ensemble.

## 4 Related work

Time series forecasting is a challenging task, and the idea of automated forecasting has long intrigued statistics and ML researchers. An early influential work on automated forecasting was the R package `forecast` (Hyndman and Khandakar, 2008) that introduced the AutoETS and AutoARIMA models. These models automatically tune their parameters (e.g., trend, seasonality) for each individual time series using an in-sample information criterion.

The following decade saw the growing focus on deep learning models for time series (Benidis et al., 2022; Wen et al., 2017; Salinas et al., 2020; Lim et al., 2021; Oreshkin et al., 2020). Several works have explored how such neural-network-based models can be combined with AutoML techniques to generate automated forecasting solutions (Van Kuppevelt et al., 2020; Shah et al., 2021; Javeri et al., 2021). Another line of research focused on optimizing the entire forecasting pipeline—including data preprocessing and feature engineering—not just hyperparameter tuning for individual models (Dahl, 2020; Kurian et al., 2021; da Silva et al., 2022). A recent survey by Meisenbacher et al. (2022) provides an overview of such automated pipelines.

Even though AutoML for forecasting is becoming an active research topic, few of the recent developments have found their way from academic papers to software packages. Available open-source AutoML forecasting libraries include AutoPyTorch–Forecasting (Deng et al., 2022), AutoTS (Catlin, 2022) and PyCaret (Ali, 2020). In contrast to these frameworks, AG–TS supports probabilistic forecasting and focuses on ease of use, allowing users to generate forecasts in a few lines of code.

## 5 Experiments

### 5.1 Setup

The goal of our experiments is to evaluate the point and probabilistic forecast accuracy of AG–TS. As baselines, we use various statistical and ML-based forecasting methods.

**Baseline methods.** **AutoARIMA**, **AutoETS**, and **AutoTheta** are established statistical forecasting models that automatically tune model parameters for each time series individually based on an information criterion (Hyndman et al., 2008). This means, such models do not require a validation set and use in-sample statistics for model tuning. **StatEnsemble** is defined by taking the median of the predictions of the three statistical models. Such statistical ensembles, despite their simplicity, have been shown to achieve competitive results in forecasting competitions (Makridakis et al., 2018). We use Python implementations of all these methods provided by the StatsForecast library (Garza et al., 2022). We additionally use **Seasonal Naive** as a sanity-check baseline that all other methods are compared against (Hyndman and Athanasopoulos, 2018).

For ML-based methods, we include two established deep learning forecasting models, **DeepAR** (Salinas et al., 2020) and **Temporal Fusion Transformer (TFT)** (Lim et al., 2021). We use the PyTorch implementations of these models provided by GluonTS (Alexandrov et al., 2020). Finally, we include the AutoML forecasting framework **AutoPyTorch–Forecasting** (Deng et al., 2022) to our comparison. AutoPyTorch builds deep learning forecasting models by combining neural architecture search (e.g., by trying various encoder modules) and hyperparameter optimization (e.g., by tuning the learning rate). The search process is powered by a combination of Bayesian and multi-fidelity optimization. Similar to AutoGluon, the models are combined using ensemble selection (Caruana et al., 2004).

**Datasets.** In our evaluation we use 29 publicly available forecasting benchmark datasets provided via GluonTS. These include datasets from the Monash Forecasting Repository (Godahewa et al., 2021), such as the M1, M3 and M4 competition data (Makridakis and Hibon, 2000; Makridakis et al., 2018). We selected the datasets from the Monash Repository that contain more than a single time series and fewer than 15M total time steps. Our selection of datasets covers various scenarios that can be encountered in practice—from small datasets (M1 and M3), to datasets with a few long time series (Electricity, Pedestrian Counts) and large collections of medium-sized time series (M4). A comprehensive list of dataset statistics are provided in Table 8 in the appendix.

**Configuration.** We train the `TimeSeriesPredictor` from AG–TS with `best_quality` presets, as these are designed to produce the most accurate forecasts, and set the `time_limit` to 4 hours. Note that the presets were fixed a priori and not optimized using the benchmark datasets. DeepAR and TFT are also trained for up to 4 hours with early stopping on validation loss with patience set to 200. For these models, the model checkpoint achieving the best validation loss is used to generate the test predictions. The time limit for AutoPyTorch is similarly set to 4 hours. We set no time limit for the remaining statistical models, as they do not support such functionality. In case the runtime of a single experiment exceeds 6 hours, the job is interrupted and the result is marked as failure. More details about the configuration are available in Appendix A.3.

All models are trained using AWS `m6i.4xlarge` cloud instances (16 vCPU cores, 64 GB RAM). We use CPU instances to fairly evaluate the CPU-only baselines, though AG–TS additionally supports GPU training. Each run is repeated 5 times using different random seeds for non-deterministic models. We run all experiments using AutoMLBenchmark (Gijsbers et al., 2022). In the supplement, we provide full configuration details and the scripts for reproducing all experiments.

### 5.2 Forecasting Accuracy

We measure the accuracy of the **point forecasts** by reporting the **mean absolute scaled error (MASE)** of all forecasting methods on all benchmark datasets. AG–TS and AutoPyTorch are trained

Table 3: Point forecast accuracy comparison of baseline methods with AutoGluon (based on the MASE metric) on 29 datasets. Listed are the number datasets where each method produced: lower error than AutoGluon (Wins), higher error (Losses), error within 0.001 (Ties), error during prediction (Failures), or the lowest error among all methods (Champion). Average rank and average error are computed using the datasets where no method failed. We rescale the errors for each dataset between [0, 1] to ensure that averaging is meaningful. The final column reports the win rate versus the Seasonal Naive baseline. Individual results are given in Table 9.

| Framework | Wins | Losses | Ties | Failures | Champion | Average rank | Average rescaled error | Win rate vs. baseline |
|---|---|---|---|---|---|---|---|---|
| AutoGluon (MASE) | - | - | - | **0** | 19 | **2.08** | **0.073** | **100.0%** |
| StatEnsemble | 6 | 20 | 0 | 3 | 3 | 3.12 | 0.238 | 82.8 % |
| AutoPyTorch (MASE) | 4 | 25 | 0 | **0** | 2 | 4.12 | 0.257 | 93.1% |
| AutoETS | 4 | 25 | 0 | **0** | 1 | 4.64 | 0.374 | 75.9 % |
| AutoTheta | 4 | 23 | 0 | 2 | 0 | 4.92 | 0.427 | 72.4 % |
| DeepAR | 4 | 24 | 0 | 1 | 2 | 5.08 | 0.434 | 93.1 % |
| AutoARIMA | 4 | 22 | 0 | 3 | 1 | 5.92 | 0.612 | 79.3 % |
| TFT | 2 | 27 | 0 | **0** | 1 | 6.12 | 0.635 | 75.9 % |

Table 4: Probabilistic forecast accuracy comparison of each baseline method with AutoGluon (based on the wQL metric) on 29 datasets. The columns are defined as in Table 3. Results for individual models and datasets are given in Table 10.

| Framework | Wins | Losses | Ties | Failures | Champion | Average rank | Average rescaled error | Win rate vs. baseline |
|---|---|---|---|---|---|---|---|---|
| AutoGluon (wQL) | - | - | - | **0** | 19 | **1.80** | **0.086** | **100.0%** |
| StatEnsemble | 3 | 23 | 0 | 3 | 0 | 3.36 | 0.330 | 86.2% |
| DeepAR | 5 | 23 | 0 | 1 | 1 | 4.08 | 0.455 | 89.7% |
| TFT | 5 | 24 | 0 | **0** | 5 | 4.24 | 0.487 | 89.7% |
| AutoETS | 3 | 26 | 0 | **0** | 2 | 4.40 | 0.489 | 69.0% |
| AutoTheta | 2 | 25 | 0 | 2 | 1 | 5.00 | 0.545 | 69.0% |
| AutoARIMA | 4 | 22 | 0 | 3 | 1 | 5.12 | 0.641 | 82.8% |

to optimize the MASE metric, while all other models are trained using their normal training procedure. We report the aggregate statistics in Table 3, and provide the full results for individual models and datasets in Table 9 in the appendix.

We measure the accuracy of the **probabilistic (quantile) forecasts** by reporting the **mean weighted quantile loss (wQL)** averaged over 9 quantile levels $q \in \{0.1, 0.2, ..., 0.9\}$. AG–TS is configured to optimize the wQL metric. We exclude AutoPyTorch from this comparison since this framework does not support probabilistic forecasting. We report the aggregate statistics in Table 4, and provide the full results for individual models and datasets in Table 10 in the appendix.

Some of the frameworks failed to generate forecasts on certain datasets. AutoARIMA, AutoTheta and StatEnsemble did not finish training on some datasets (Electricity–Hourly, KDD Cup 2018, and Pedestrian Counts) within 6 hours. This is caused by the poor scaling of these models to very long time series. DeepAR model fails on one dataset (Web Traffic Weekly) due to numerical errors encountered during training.

**Discussion**. The results demonstrate that AG–TS outperforms all other frameworks, achieving the best average rank and rescaled error for both point and probabilistic forecasts, and even beating the best-in-hindsight competing method on 19 out of 29 datasets.

StatEnsemble places second after AG–TS. The statistical ensemble performs especially well on small datasets such as M1 and M3. This demonstrates that in the low-data regime simple approaches,

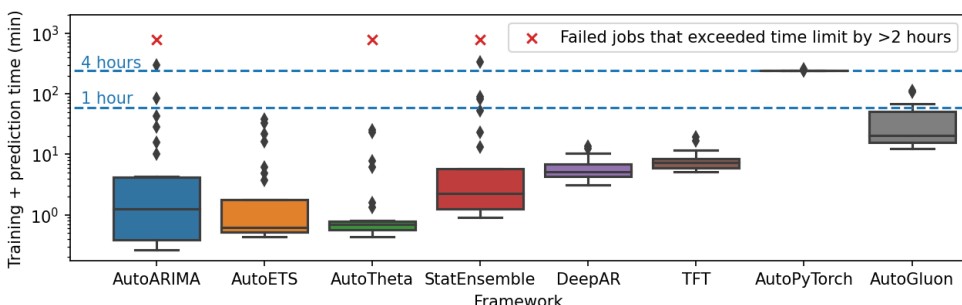

Figure 2: Total runtime of each framework across all datasets. AutoGluon always completes training and prediction under the time limit and achieves a mean runtime of 33 minutes. AutoPyTorch is always trained for the full 4 hour time limit. Statistical models train faster in most cases, but may take an extremely long time to train on datasets with long time series. The runtimes for individual models and datasets are provided in Table 11.

like ensembling by taking the median, may perform better than the learned ensemble selection strategy employed by both AutoML frameworks.

AutoPyTorch achieves similar performance to StatEnsemble in point forecasting across most performance indicators. Interestingly, AG–TS tends to outperform AutoPyTorch on larger datasets like M4. This means that AG–TS's strategy of training various light-weight models performs well in this setting under the limited time budget. Also note, configuring AutoPyTorch requires more code and domain knowledge, compared to the 3 lines of code necessary to reproduce the above results by AG–TS.

Deep learning models DeepAR and TFT perform well in terms of probabilistic forecasting, but fall behind simple statistical approaches in point forecasts. This makes sense, since the objective functions optimized by these deep learning models are designed for probabilistic forecasting.

## 5.3 Runtime Comparison

High accuracy is not the only important property of an AutoML system—the ability to generate predictions in a reasonable amount of time is often necessary in practice. To evaluate the efficiency of AG–TS, we compare its runtime with other frameworks. We visualize the runtime of each framework across all datasets in Figure 2. Note that here we compare the total runtime defined as the sum of training and prediction times. This reflects the typical forecasting workflow in practice, where the forecast is generated once for each time series. Moreover, it's hard to distinguish between the training and prediction time for local models, where a new model is trained for each new time series.

AG–TS completes training and prediction under the 4-hour time limit for all 29 datasets, and achieves mean runtime of 33 minutes. While statistical models are faster on average, they can be extremely slow to train on datasets consisting of long time series. For instance, the runtimes of AutoARIMA, AutoTheta and StatEnsemble exceed 6 hours for 3 datasets with long time series. The deep learning models DeepAR and TFT have higher median runtime compared to the statistical models, but never reach the 4 hour time limit due to early stopping. Finally, AutoPyTorch always consumes the entire 4 hour time budget due to its design.

To summarize, AG–TS is able to produce accurate forecasts under mild time budgets. While, on average, AG–TS takes more time than the individual models, it produces more accurate forecasts and avoids the extremely long runtimes sometimes exhibited by local models. The results also demonstrate that limited training time is better spent training and ensembling many diverse models (as done by AG–TS), rather than hyperparameter tuning a restricted set of models (as done by AutoPyTorch).

Table 5: Ablation study. We compare the point forecast accuracy of AutoGluon, where certain component models are removed, ensembling is disabled, or the time limit is reduced. All versions except AutoGluon-1h and AutoGluon-10m are trained for 4 hours. The columns are defined and the scores are computed as in Table 3.

| Framework | Champion | Average rank | Average rescaled error |
|---|---|---|---|
| AutoGluon-1h | **19** | **2.04** | **0.070** |
| AutoGluon-4h | **19** | 2.08 | 0.073 |
| NoStatModels | 16 | 2.12 | 0.094 |
| NoTabularModels | 15 | 2.12 | 0.085 |
| NoDeepModels | 15 | 2.28 | 0.124 |
| AutoGluon-10m | 14 | 2.50 | 0.099 |
| NoEnsemble | 7 | 3.52 | 0.177 |

## 5.4 Ablations

Finally, we perform ablations to understand the effect of different components on the final performance. We compare the point forecast accuracy of the `TimeSeriesPredictor` trained for 4 hours with MASE evalauation metric (Section 5.2) against several variations with certain disabled components. First, we exclude some base models from the presets: statistical models (**NoStatModels**), deep learning models (**NoDeepModels**), and tabular models (**NoTabularModels**). We also consider reducing the time limit to 1 hour (**AutoGluon-1h**) or 10 minutes (**AutoGluon-10m**), as well disabling the final ensembling step (**NoEnsemble**). In the latter case, AG–TS predicts using the model with the best validation score. The rest of the setup is identical to Section 5.2.

Table 5 shows the metrics for the different model variations, each compared to the baselines from Section 5.2. AutoGluon-4h and AutoGluon-1h produce nearly identical results. This is not surprising, as the 4-hour version finishes training under 1 hour for most datasets (Figure 2). Interestingly, AutoGluon achieves strong results even with a 10-minute time limit, achieving the best average rank and outperforming the best-in-hindsight model on 14 out of 29 datasets.

Removing the ensembling step has the most detrimental effect on the overall accuracy. This highlights the importance of ensembling, confirming the findings of other works (Makridakis et al., 2018; Borchert et al., 2022). The ablations also show that all 3 classes of models used by AutoGluon are important for the overall performance, deep learning models being the most critical component.

## 6 Future Work

Our experiments demonstrate the strong forecasting accuracy achieved by AG–TS. Despite these encouraging initial results, we aim to continue developing the library, adding new functionality and further boost the forecasting performance. This includes incorporating the various ideas in the space of AutoML for forecasting (Meisenbacher et al., 2022), with focus on the following directions.

**Ensembling**. Advanced ensembling strategies, such as stacking (Ting and Witten, 1997), lie at the core of modern high-performing AutoML systems (Erickson et al., 2020). How to best generalize these techniques to probabilistic forecasting is an active, but still open research question (Gastinger et al., 2021; Wang et al., 2022).

**Calibration**. Many practical tasks require guarantees on the uncertainty estimates associated with the forecasts. Conformal prediction methods (Stankeviciute et al., 2021; Xu and Xie, 2021) provide one way to obtain such guarantees, and we plan to incorporate them into AG–TS in the future.

**New problem types**. AG–TS supports the most common types of forecasting tasks, such as probabilistic forecasting or handling covariates. However, there are several settings that are currently (as

of v0.8) not supported. These include so-called cold-start forecasting (where little historic data is available) and generating forecast explanations (Rojat et al., 2021). Another interesting potential application for AG–TS is assisting judgemental forecasting. In this context, AG–TS could serve as a "tool" queried by a large language model (LLM) (Schick et al., 2023) to generate qualitative forecasts. More generally, combinations of LLM with AutoML frameworks are an exciting direction for future work (Tornede et al., 2023).

**Scalability**. In our experiments we consider datasets with up to $\approx 10^7$ time steps across all time series. Modern applications, however, sometimes require operating on even larger scales. This would require improving efficiency of existing models and developing new efficient AutoML techniques.

## 7 Conclusions

In this work, we introduced AutoGluon–TimeSeries, a powerful and user-friendly open-source AutoML library for probabilistic time series forecasting. By combining statistical models and deep learning forecasting approaches with ensembling techniques, AutoGluon–TimeSeries is able to achieve strong empirical results on a range of benchmark datasets. With the ability to generate accurate point and quantile forecasts with just 3 lines of Python code, this framework is poised to make time series forecasting more accessible and efficient for a wide range of users.

## 8 Broader Impact Statement

AutoGluon–TimeSeries enables users to generate accurate forecasts in a few lines of code. This democratizes machine learning, lowering the barrier to entry to forecasting for non-experts. At the same time, AutoGluon–TimeSeries can be used by experienced users to design highly accurate forecasting pipelines. More accurate forecasts can directly translate to real-world impact in various domains. For example, forecasting renewable energy generation is a crucial component of smart grid management (Tripathy and Prusty, 2021); accurately predicting demand leads to more efficient inventory management and increased revenue (Makridakis et al., 2022).

The potential negative impacts of the proposed approach are similar to those of other forecasting models. One such danger arises when the limitations of forecasting methods are not taken into account in the context of decision making (e.g., when guiding policy decisions). As forecasting models only capture statistical dependencies, they may be misleading when trying to estimate effects of actions or interventions.

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

## A  Supplementary Materials

### A.1  Evaluation Metrics

**MASE.** Mean absolute scaled error is the standard metric for evaluating the accuracy of point forecasts.

$$\text{MASE} = \frac{1}{N} \sum_{i=1}^{N} \frac{1}{H} \frac{\sum_{h=1}^{H} |y_{i,T+h} - \hat{y}_{i,T+h}|}{\sum_{t=1}^{T-s} |y_{i,t+s} - y_{i,t}|}$$

MASE is scale-invariant and does not suffer from the limitations of other metrics, such as being undefined when the target time series equals zero (Hyndman and Athanasopoulos, 2018). We compute the metric using the median (0.5 quantile) forecast produced by each model.

**wQL.** Weighted quantile loss for a single quantile level $q$ is defined as

$$\text{wQL}[q] = 2 \frac{\sum_{i=1}^{N} \sum_{h=1}^{H} \left[ q \cdot \max(y_{i,T+h} - \hat{y}_{i,T+h}^{q}, 0) + (1-q) \cdot \max(\hat{y}_{i,T+h}^{q} - y_{i,T+h}, 0) \right]}{\sum_{i=1}^{N} \sum_{h=1}^{H} |y_{i,T+h}|}$$

In our experiments, we report the mean wQL averaged over 9 quantile levels $\mathcal{Q} = \{0.1, 0.2, ..., 0.9\}$.

$$\text{wQL} = \frac{1}{|\mathcal{Q}|} \sum_{q \in \mathcal{Q}} \text{wQL}[q]$$

### A.2  Reproducibility

We ran all experiments using AutoMLBenchmark (Gijsbers et al., 2022). We provide a fork of AMLB that includes all scripts necessary to reproduce the results from our paper in the following GitHub repository `https://github.com/shchur/automlbenchmark/tree/autogluon-timeseries-automl23/autogluon_timeseries_automl23`.

### A.3  Model Configuration

We trained the baseline models DeepAR, TFT, AutoARIMA, AutoETS, AutoTheta with the default hyperparameter configurations provided by the respective libraries. For DeepAR and TFT, the last `prediction_length` time steps of each time series were reserved as a validation set. Both models were trained for the full duration of 4 hours, saving the parameters and evaluating the validation loss at each epoch. The parameters achieving the lowest validation loss were then used for prediction. No HPO was performed for these two models, as AutoPyTorch already trains similar deep learning models with HPO.

For AutoPyTorch, we used the reference implementation by the authors.[3] We set the target metric to "mean_MASE_forecasting", budget_type="epochs", min_budget=5, max_budget=50, and resampling_strategy=HoldoutValTypes.time_series_hold_out_validation. We also set torch_num_threads to 16 (the number of vCPU cores).

In our experiments, we used AG–TS v0.8.2, the latest release at the time of publication. We used the "best_quality" presets and set `eval_metric` to either "MASE" or "mean_wQuantileLoss", depending on the experiment. All other parameters of the `TimeSeriesPredictor` were set to their default values. The "best_quality" presets include the following models: AutoETS, AutoARIMA, Theta (from StatsForecast), DeepAR, PatchTST, TFT (from GluonTS), DirectTabular, RecursiveTabular (wrappers around AutoGluon–Tabular and MLForecast), plus the baseline methods Naive and SeasonalNaive. The non-default hyperparameters of the individual models used by the `best_quality` presets are provided in Table 6.

---

[3] `https://github.com/dengdifan/Auto-PyTorch/blob/ecml22_apt_ts/examples/APT-TS/APT_task.py`

The guiding principle for developing the presets for AG–TS can be summarized as "keep defaults whenever possible, except the cases where the defaults are clearly suboptimal". For example, we set `allowmean=True` for AutoARIMA to allow this model to handle time series with non-zero mean. For deep learning models, we increase the batch size from 32 to 64 since larger batch sizes typically lead to faster convergence for all deep learning models. The `context_length` is capped at a minimum value because the default setting `context_length=prediction_length` can result in models that ignore most of the history if `prediction_length` is very short. For PatchTST, we set the `context_length` to the value used in the respective publication (Nie et al., 2023).

The versions of frameworks used in our experiments are listed in Table 7.

Table 6: Non-default hyperparameters that AutoGluon sets for the underlying models. The remaining parameters are all set to their defaults in the respective libraries. Models not listed here (Naive, SeasonalNaive, AutoETS, DirectTabular, Theta) have all their hyperparameters set to the default values.

| Model | Hyperparameter | Value |
|---|---|---|
| AutoARIMA | `allowmean` | `True` |
| | `approximation` | `True` |
| DeepAR | `batch_size` | `64` |
| | `context_length` | `max(10, 2 * prediction_length)` |
| | `num_samples` | `250` |
| PatchTST | `batch_size` | `64` |
| | `context_length` | `96` |
| TFT | `batch_size` | `64` |
| | `context_length` | `max(64, 2 * prediction_length)` |
| RecursiveTabular | `tabular_hyperparameters` | `{"GBM", "NN_TORCH"}` |

Table 7: Versions of the frameworks used during evaluation.

| Framework | Version |
|---|---|
| AutoGluon | 0.8.2 |
| AutoPyTorch | 0.2.1 |
| GluonTS | 0.13.2 |
| MLForecast | 0.7.3 |
| StatsForecast | 1.5.0 |
| Python | 3.9 |
| PyTorch | 1.13.1+cpu |

Table 8: Statistics of the benchmark datasets used in our experimental evaluation. Frequency is represented by pandas offset aliases. Seasonality depends on the frequency, and is used to configure statistical models and compute the MASE metric.

| Dataset | # series | # time steps | Prediction length | Frequency | Seasonality |
|---|---|---|---|---|---|
| Car Parts | 2,674 | 104,286 | 12 | M | 12 |
| CIF 2016 | 72 | 6,244 | 12 | M | 12 |
| COVID | 266 | 48,412 | 30 | D | 7 |
| Electricity Hourly | 321 | 8,428,176 | 48 | H | 24 |
| Electricity Weekly | 321 | 47,508 | 8 | W | 1 |
| FRED-MD | 107 | 76,612 | 12 | M | 12 |
| Hospital | 767 | 55,224 | 12 | M | 12 |
| KDD Cup 2018 | 270 | 2,929,404 | 48 | H | 24 |
| M1 Monthly | 617 | 44,892 | 18 | M | 12 |
| M1 Quarterly | 203 | 8,320 | 8 | Q | 4 |
| M1 Yearly | 181 | 3,429 | 6 | Y | 1 |
| M3 Monthly | 1,428 | 141,858 | 18 | M | 12 |
| M3 Other | 174 | 11,933 | 8 | Q | 1 |
| M3 Quarterly | 756 | 30,956 | 8 | Q | 4 |
| M3 Yearly | 645 | 14,449 | 6 | Y | 1 |
| M4 Daily | 4,227 | 9,964,658 | 14 | D | 7 |
| M4 Hourly | 414 | 353,500 | 48 | H | 24 |
| M4 Monthly | 48,000 | 10,382,411 | 18 | M | 12 |
| M4 Quarterly | 24,000 | 2,214,108 | 8 | Q | 4 |
| M4 Weekly | 359 | 366,912 | 13 | W | 1 |
| M4 Yearly | 22,974 | 707,265 | 6 | Y | 1 |
| NN5 Daily | 111 | 81,585 | 56 | D | 7 |
| NN5 Weekly | 111 | 11,655 | 8 | W | 1 |
| Pedestrian Counts | 66 | 3,129,178 | 48 | H | 24 |
| Tourism Monthly | 366 | 100,496 | 24 | M | 12 |
| Tourism Quarterly | 427 | 39,128 | 8 | Q | 4 |
| Tourism Yearly | 518 | 10,685 | 4 | Y | 1 |
| Vehicle Trips | 262 | 45,253 | 7 | D | 7 |
| Web Traffic Weekly | 145,063 | 15,376,678 | 8 | W | 1 |

Table 9: Point forecast accuracy, as measured by MASE (lower is better). For non-deterministic methods (DeepAR, TFT, AutoPyTorch, AutoGluon) we report the mean and standard deviation of the scores computed over 5 random seeds. "d.n.f" denotes cases where a method did not generate a forecast in 6 hours. "N/A" denotes model failure.

| | SeasonalNaive | AutoARIMA | AutoETS | AutoTheta | StatEnsemble | DeepAR | TFT | AutoPyTorch | AutoGluon |
|---|---|---|---|---|---|---|---|---|---|
| Car Parts | 1.127 | 1.118 | 1.133 | 1.208 | 1.052 | 0.749 (0.001) | 0.751 (0.002) | **0.746** (0.0) | 0.747 (0.0) |
| CIF 2016 | 1.289 | 1.069 | **0.898** | 1.006 | 0.945 | 1.278 (0.088) | 1.372 (0.085) | 1.023 (0.069) | 1.073 (0.006) |
| COVID | 8.977 | 6.029 | 5.907 | 7.719 | 5.884 | 7.166 (0.334) | 5.192 (0.211) | **4.911** (0.086) | 5.805 (0.0) |
| Electricity Hourly | 1.405 | d.n.f. | 1.465 | d.n.f. | d.n.f. | 1.251 (0.006) | 1.389 (0.025) | 1.420 (0.123) | 1.227 (0.003) |
| Electricity Weekly | 3.037 | 3.009 | 3.076 | 3.113 | 3.077 | 2.447 (0.211) | 2.861 (0.122) | 2.322 (0.277) | 1.892 (0.0) |
| FRED-MD | 1.101 | **0.478** | 0.505 | 0.564 | 0.498 | 0.634 (0.038) | 0.901 (0.086) | 0.682 (0.058) | 0.656 (0.0) |
| Hospital | 0.921 | 0.820 | 0.766 | 0.764 | 0.753 | 0.771 (0.008) | 0.814 (0.012) | 0.770 (0.003) | **0.741** (0.001) |
| KDD Cup 2018 | 0.975 | d.n.f. | 0.988 | 1.010 | d.n.f. | 0.841 (0.036) | 0.844 (0.065) | 0.764 (0.047) | **0.709** (0.026) |
| M1 Monthly | 1.314 | 1.152 | 1.083 | 1.092 | **1.045** | 1.117 (0.029) | 1.534 (0.063) | 1.278 (0.115) | 1.235 (0.001) |
| M1 Quarterly | 2.078 | 1.770 | 1.665 | 1.667 | 1.622 | 1.742 (0.028) | 2.099 (0.108) | 1.813 (0.056) | 1.615 (0.0) |
| M1 Yearly | 4.894 | 3.870 | 3.950 | 3.659 | 3.769 | 3.674 (0.161) | 4.318 (0.122) | 3.407 (0.078) | 3.371 (0.007) |
| M3 Monthly | 1.146 | 0.934 | 0.867 | 0.855 | 0.845 | 0.960 (0.017) | 1.062 (0.04) | 0.956 (0.083) | **0.822** (0.0) |
| M3 Other | 3.089 | 2.245 | 1.801 | 2.009 | **1.769** | 2.061 (0.182) | 1.926 (0.028) | 1.871 (0.024) | 1.837 (0.004) |
| M3 Quarterly | 1.425 | 1.419 | 1.121 | 1.119 | 1.096 | 1.198 (0.037) | 1.176 (0.036) | 1.180 (0.032) | 1.057 (0.002) |
| M3 Yearly | 3.172 | 3.159 | 2.695 | 2.608 | 2.627 | 2.694 (0.096) | 2.818 (0.019) | 2.691 (0.026) | 2.520 (0.002) |
| M4 Daily | 1.452 | 1.153 | 1.228 | 1.149 | 1.145 | 1.145 (0.026) | 1.176 (0.018) | 1.152 (0.009) | 1.156 (0.0) |
| M4 Hourly | 1.193 | 1.029 | 1.609 | 2.456 | 1.157 | 1.484 (0.151) | 3.391 (0.442) | 1.345 (0.404) | **0.807** (0.001) |
| M4 Monthly | 1.079 | 0.812 | 0.803 | 0.834 | 0.780 | 0.933 (0.01) | 0.947 (0.005) | 0.851 (0.025) | 0.782 (0.0) |
| M4 Quarterly | 1.602 | 1.276 | 1.167 | 1.183 | 1.148 | 1.367 (0.171) | 1.277 (0.015) | 1.176 (0.022) | 1.139 (0.0) |
| M4 Weekly | 2.777 | 2.355 | 2.548 | 2.608 | 2.375 | 2.418 (0.026) | 2.625 (0.038) | 2.369 (0.177) | **2.035** (0.001) |
| M4 Yearly | 3.966 | 3.720 | 3.077 | 3.085 | 3.032 | 3.858 (0.694) | 3.220 (0.097) | 3.093 (0.041) | 3.019 (0.001) |
| NN5 Daily | 1.011 | 0.935 | 0.870 | 0.878 | 0.859 | 0.812 (0.01) | 0.789 (0.004) | 0.807 (0.021) | **0.761** (0.004) |
| NN5 Weekly | 1.063 | 0.998 | 0.980 | 0.963 | 0.977 | 0.915 (0.085) | 0.884 (0.012) | 0.865 (0.025) | 0.860 (0.0) |
| Pedestrian Counts | 0.369 | d.n.f. | 0.553 | d.n.f. | d.n.f. | 0.309 (0.005) | 0.373 (0.01) | 0.354 (0.024) | 0.312 (0.009) |
| Tourism Monthly | 1.631 | 1.585 | 1.529 | 1.666 | 1.469 | 1.461 (0.025) | 1.719 (0.08) | 1.495 (0.009) | 1.442 (0.0) |
| Tourism Quarterly | 1.699 | 1.655 | 1.578 | 1.648 | 1.539 | 1.599 (0.062) | 1.830 (0.047) | 1.647 (0.034) | 1.537 (0.002) |
| Tourism Yearly | 3.552 | 4.044 | 3.183 | 2.992 | 3.231 | 3.476 (0.165) | 2.916 (0.197) | 3.004 (0.053) | 2.946 (0.007) |
| Vehicle Trips | 1.302 | 1.427 | 1.301 | 1.284 | 1.203 | 1.162 (0.016) | 1.227 (0.02) | 1.162 (0.019) | 1.113 (0.0) |
| Web Traffic Weekly | 1.066 | 1.189 | 1.207 | 1.108 | 1.068 | N/A | 0.973 (0.022) | 0.962 (0.01) | 0.938 (0.0) |

Table 10: Probabilistic forecast accuracy, as measured by wQL (lower is better). For non-deterministic methods (DeepAR, TFT, AutoGluon) we report the mean and standard deviation of the scores computed over 5 random seeds. "d.n.f." denotes cases where a method did not generate a forecast in 6 hours. "N/A" denotes model failure.

| | SeasonalNaive | AutoARIMA | AutoETS | AutoTheta | StatEnsemble | DeepAR | TFT | AutoGluon |
|---|---|---|---|---|---|---|---|---|
| Car Parts | 1.717 | 1.589 | 1.338 | 1.367 | 1.324 | 0.963 (0.009) | **0.878** (0.004) | 0.923 (0.0) |
| CIF 2016 | 0.031 | 0.017 | 0.039 | 0.027 | 0.028 | 0.114 (0.024) | **0.010** (0.002) | 0.019 (0.0) |
| COVID | 0.140 | **0.030** | 0.046 | 0.094 | 0.046 | 0.072 (0.02) | 0.031 (0.003) | **0.030** (0.0) |
| Electricity Hourly | 0.108 | d.n.f. | 0.100 | d.n.f. | d.n.f. | 0.081 (0.002) | 0.097 (0.001) | **0.076** (0.0) |
| Electricity Weekly | 0.141 | 0.138 | 0.144 | 0.146 | 0.141 | 0.123 (0.041) | 0.118 (0.011) | 0.088 (0.0) |
| FRED-MD | 0.104 | 0.056 | **0.050** | 0.057 | 0.054 | 0.054 (0.021) | 0.114 (0.011) | 0.056 (0.0) |
| Hospital | 0.062 | 0.058 | 0.053 | 0.055 | 0.053 | 0.053 (0.001) | 0.054 (0.001) | **0.051** (0.0) |
| KDD Cup 2018 | 0.489 | d.n.f. | 0.550 | 0.553 | d.n.f. | 0.363 (0.014) | 0.488 (0.054) | **0.323** (0.014) |
| M1 Monthly | 0.153 | 0.146 | 0.163 | 0.159 | 0.152 | 0.136 (0.008) | 0.224 (0.016) | **0.135** (0.0) |
| M1 Quarterly | 0.119 | 0.088 | 0.081 | 0.082 | 0.083 | 0.084 (0.003) | 0.093 (0.006) | 0.090 (0.0) |
| M1 Yearly | 0.184 | 0.160 | 0.139 | 0.137 | 0.142 | 0.142 (0.029) | 0.127 (0.004) | 0.134 (0.001) |
| M3 Monthly | 0.124 | 0.102 | 0.093 | 0.095 | 0.092 | 0.098 (0.001) | 0.109 (0.003) | **0.089** (0.0) |
| M3 Other | 0.047 | 0.035 | 0.032 | 0.035 | **0.031** | 0.036 (0.002) | 0.033 (0.001) | **0.031** (0.0) |
| M3 Quarterly | 0.083 | 0.079 | 0.069 | 0.070 | 0.068 | 0.073 (0.001) | 0.071 (0.001) | **0.065** (0.0) |
| M3 Yearly | 0.141 | 0.162 | 0.129 | 0.128 | 0.128 | 0.117 (0.002) | 0.133 (0.001) | **0.114** (0.0) |
| M4 Daily | 0.030 | 0.023 | 0.025 | 0.023 | 0.023 | 0.023 (0.0) | 0.023 (0.0) | **0.022** (0.0) |
| M4 Hourly | 0.039 | 0.036 | 0.070 | 0.041 | 0.037 | 0.065 (0.03) | 0.038 (0.002) | **0.030** (0.001) |
| M4 Monthly | 0.109 | 0.085 | 0.085 | 0.088 | 0.082 | 0.092 (0.003) | 0.089 (0.001) | **0.081** (0.0) |
| M4 Quarterly | 0.099 | 0.082 | 0.079 | 0.079 | 0.076 | 0.084 (0.005) | 0.083 (0.001) | **0.075** (0.0) |
| M4 Weekly | 0.073 | 0.050 | 0.052 | 0.053 | 0.050 | 0.046 (0.001) | 0.049 (0.001) | **0.041** (0.0) |
| M4 Yearly | 0.138 | 0.130 | 0.111 | 0.115 | 0.109 | 0.124 (0.006) | 0.116 (0.004) | **0.104** (0.0) |
| NN5 Daily | 0.292 | 0.169 | 0.162 | 0.188 | 0.164 | 0.148 (0.002) | 0.145 (0.001) | **0.140** (0.0) |
| NN5 Weekly | 0.142 | 0.090 | 0.088 | 0.090 | 0.089 | 0.084 (0.007) | 0.085 (0.001) | **0.078** (0.0) |
| Pedestrian Counts | 0.675 | d.n.f. | 0.764 | d.n.f. | d.n.f. | **0.230** (0.006) | 0.261 (0.008) | 0.238 (0.013) |
| Tourism Monthly | 0.088 | 0.095 | 0.101 | 0.091 | 0.085 | 0.086 (0.005) | 0.103 (0.01) | **0.083** (0.0) |
| Tourism Quarterly | 0.099 | 0.098 | 0.070 | **0.061** | 0.070 | 0.068 (0.002) | 0.083 (0.005) | 0.072 (0.0) |
| Tourism Yearly | 0.170 | 0.156 | 0.157 | 0.176 | 0.155 | 0.141 (0.016) | **0.102** (0.006) | 0.152 (0.0) |
| Vehicle Trips | 0.112 | 0.100 | 0.115 | 0.120 | 0.103 | 0.090 (0.002) | 0.099 (0.005) | 0.087 (0.0) |
| Web Traffic Weekly | 0.936 | 0.475 | $8 \cdot 10^{13}$ | 0.503 | 0.474 | N/A | **0.223** (0.011) | 0.225 (0.0) |

Table 11: Average run time of each method (in minutes).

| Dataset | SeasonalNaive | AutoARIMA | AutoETS | AutoTheta | StatEnsemble | DeepAR | TFT | AutoPyTorch | AutoGluon |
|---|---|---|---|---|---|---|---|---|---|
| Car Parts | 0.1 | 2.4 | 0.6 | 0.7 | 3.3 | 6.9 | 9.2 | 240.3 | 17.4 |
| CIF 2016 | 0.1 | 0.4 | 0.5 | 0.6 | 1.3 | 4.1 | 6.2 | 240.2 | 16.7 |
| COVID | 0.1 | 1.4 | 0.5 | 0.7 | 2.3 | 7.9 | 8.8 | 240.4 | 29.3 |
| Electricity Hourly | 0.2 | >360 | 21.6 | >360 | >360 | 10.4 | 19.5 | 240.4 | 61.2 |
| Electricity Weekly | 0.2 | 0.3 | 0.4 | 0.5 | 1.0 | 3.1 | 6.6 | 240.2 | 14.9 |
| FRED-MD | 0.1 | 2.4 | 0.7 | 0.6 | 3.4 | 6.8 | 5.5 | 240.2 | 16.8 |
| Hospital | 0.1 | 0.9 | 0.7 | 0.7 | 2.1 | 4.6 | 7.6 | 240.2 | 17.4 |
| KDD Cup 2018 | 0.1 | >360 | 16.3 | 22.8 | >360 | 12.4 | 11.9 | 240.3 | 56.0 |
| M1 Monthly | 0.1 | 1.5 | 0.8 | 0.7 | 2.7 | 5.5 | 6.2 | 240.2 | 21.6 |
| M1 Quarterly | 0.1 | 0.3 | 0.5 | 0.7 | 1.3 | 5.9 | 5.4 | 240.2 | 15.6 |
| M1 Yearly | 0.1 | 0.3 | 0.4 | 0.4 | 0.9 | 4.2 | 5.2 | 240.2 | 12.9 |
| M3 Monthly | 0.1 | 4.0 | 1.0 | 0.8 | 5.8 | 5.1 | 5.9 | 240.3 | 24.2 |
| M3 Other | 0.1 | 0.3 | 0.4 | 0.4 | 0.9 | 5.0 | 6.0 | 240.2 | 13.6 |
| M3 Quarterly | 0.1 | 0.5 | 0.6 | 0.7 | 1.6 | 4.6 | 6.0 | 240.3 | 15.7 |
| M3 Yearly | 0.1 | 0.4 | 0.5 | 0.4 | 1.0 | 5.9 | 5.4 | 240.2 | 12.7 |
| M4 Daily | 0.2 | 28.5 | 33.0 | 25.3 | 82.3 | 6.8 | 8.4 | 240.3 | 68.7 |
| M4 Hourly | 0.1 | 84.9 | 1.8 | 0.8 | 89.5 | 9.2 | 10.9 | 240.2 | 51.2 |
| M4 Monthly | 0.3 | 296.0 | 37.6 | 7.7 | 340.3 | 4.9 | 7.9 | 242.0 | 112.1 |
| M4 Quarterly | 0.2 | 15.7 | 6.2 | 1.6 | 23.2 | 4.7 | 7.6 | 240.9 | 62.3 |
| M4 Weekly | 0.1 | 0.6 | 0.5 | 1.3 | 2.2 | 5.6 | 7.8 | 240.3 | 20.8 |
| M4 Yearly | 0.2 | 4.3 | 0.8 | 0.7 | 5.6 | 4.2 | 6.1 | 240.8 | 35.6 |
| NN5 Daily | 0.1 | 2.5 | 0.5 | 0.6 | 3.3 | 7.3 | 10.9 | 240.3 | 37.4 |
| NN5 Weekly | 0.1 | 0.3 | 0.4 | 0.4 | 1.0 | 3.6 | 6.4 | 240.2 | 13.7 |
| Pedestrian Counts | 0.1 | >360 | 4.9 | >360 | >360 | 13.5 | 16.7 | 240.7 | 56.4 |
| Tourism Monthly | 0.1 | 10.2 | 0.8 | 0.7 | 13.1 | 4.4 | 7.6 | 240.2 | 26.0 |
| Tourism Quarterly | 0.1 | 0.9 | 0.6 | 0.7 | 1.8 | 3.6 | 6.3 | 240.2 | 14.6 |
| Tourism Yearly | 0.1 | 0.3 | 0.4 | 0.4 | 1.0 | 3.5 | 5.8 | 240.3 | 12.4 |
| Vehicle Trips | 0.1 | 1.1 | 0.6 | 0.7 | 2.2 | 5.1 | 7.3 | 240.2 | 16.0 |
| Web Traffic Weekly | 0.2 | 42.3 | 3.7 | 6.2 | 52.8 | N/A | 8.3 | 260.5 | 106.0 |

