# OpenReview forum: "AutoGluon–TimeSeries: AutoML for Probabilistic Time Series Forecasting"
_automl.cc/AutoML/2023/ABCD_Track — AutoML 2023 (ABCD Track)_

### Official Review · Reviewer_xTbR · 2023-04-28

**Potential Impact On The Field Of Automl Rating:** 4
**Technical Quality And Correctness Rating:** 3
**Clarity Rating:** 4

**Summary Of Contributions:**

This paper introduces a new algorithm for forecasting time series, called AutoGluon–TimeSeries (AG–TS). This algorithm is capable of predicting how a provided times series continues in two different ways: (1) by providing single points for each new time point of the forecast, and (2) by providing a probabilistic forecast, where the algorithm provides the user for each new time point of the forecast with the values for a specified range of quantiles (say, from 0.1 to 0.9 in steps of 0.1) that determine the likelihood of a new data point to lie within this range.

AG–TS provides a flexible interface with many reasonable default values such that it can be used with very few lines of code and without any expert knowledge about the domain of the time series it is being applied to.

The authors test this approach against various existing approaches, both in terms of solution quality and running time.

**Actions Required To Increase Overall Recommendation:**

My points that I mention in “Clarity” should be addressed. Further, the authors should at the very least comment in the paper on why they believe that three iterations of the experiments for the approaches with random models is a sufficiently large number.

**Clarity:**

The paper is very well structured and written. I found almost no typos and have almost no ideas for how to improve the presentation, which is very rare. Congratulations to the authors! My only somewhat major complaint is that I am missing a detailed discussion of related work. The introduction spends very little time on explaining the setting and what approaches exist. In Section 4.1, the competing approaches are briefly explained, but I am lacking a solid overview of the state of the art. This paper contains some hints, but the full picture is not entirely clear to me. I would like to see a more detailed discussion somewhere.

Minor comments:

- In the second paragraph of the introduction (page 1), there is an incorrect comma immediately following “regression tasks for tabular data”.
- In the first sentence of the third paragraph of the introduction (page 1), the term “AutoGluon–TimeSeries” appears inside of parentheses instead of its abbreviation.
- Although the paper uses (and sort of introduces) the abbreviation AG–TS, it is not always used. This is strange to me. Since the normal name is rather lengthy and does not roll off the tongue nicely, I would prefer using the (still slightly lengthy) abbreviation as much as possible.
- The second paragraph of Section 1 contains the phrase “be known into the future”, where “into” should be just “in”.
- The display formulas have numbers that are never used. Hence, the numbers should be omitted.
- The code presented at the beginning of Section 3 (page 3) does not say that it is Python code. In fact, the paper almost never says that AG–TS is a Python framework (except for the abstract and the conclusion). It should be at least also mentioned in the introduction.
- The caption of Table 1 (page 3) says “row contains unique id”, which should be “row contains the unique ID”.
- In the paragraph about defining tasks in Section 3 (page 3), the first full sentence should either start with “The user” of with “Users”.
- The last paragraph before Section 3.2 begins (page 4) has a sentence start with “AG–TS API”. This sentence should start with “The” instead.
- The sentence that introduces equation (3) (page 5) mentions “The ensemble of the ensemble”, which makes little sense to me. I thought we only had a single ensemble? What does the meta ensemble supposedly do? Equation (3) shows me what I would have believed a single ensemble (of time series) does.
- The section in Section 4.1 titled “Baseline methods” (page 5) is missing a serial comma (which the authors seem to use) in its first sentence (after “AutoETS”).
- After the first sentence of the second paragraph of Section A.2 (page 15), there is a superfluous space before the footnote.

**Overall Review:**

This paper presents a very nice and potentially very impactful framework for forecasting time series. The framework is easy to use, and its performance against competing approaches is very good, as showcased by thoroughly executed experiments. My only concerns are that related work is not very well discussed and that some of the experiments are repeated a seemingly very small number of times. This makes it a bit harder to properly assess and appreciate the contribution of this paper.

**Potential Impact On The Field Of Automl:**

Forecasting time series is a very important and especially challenging topic with various useful applications in the real world. Hence, providing good tools that relieve users from making complicated decisions when trying to compute such forecasts is a noble goal. If such a tool manages to work almost out-of-the-box, the domain of application could be potentially drastically increased, as non-experts could also achieve good results with little effort. In addition, such a contribution has the potential to solidly advance the area of AutoML applications on temporal data.

**Reproducibility (Optional):**

The authors provide supplementary material that contains all of the necessary files and information in order to reproduce the results. While I did not test it out, I looked at the instructions and some of the files, and the quality of the material looks very good to me.

**Review Confidence:**

3: You are fairly confident in your assessment. It is possible that you did not understand some parts of the submission or that you are unfamiliar with some pieces of related work.

**Review Rating:**

9: Strong Accept: Technically flawless submission with major impact, with no obvious flaws. Should be highlighted in the program.

**Review Summary:**

This paper is very well written and presents a very nice piece of code. The tests appear sound to me, and the performance of the new framework is very good (and better than the competition). Hence, I highly recommend to accept this paper.

EDIT: The authors responded to my suggestions satisfyingly. Hence, I do not decrease my score. However, I also do not increase it, as the current score is already very high and I do not think that the result is strong enough to receive a 10.

**Technical Quality And Correctness:**

The presented algorithm (AG–TS) appears to be of very high quality. Its source code is freely available on GitHub, and the submission comes with supplementary material that explains how to reproduce the experiments that were run. The evaluation of AG–TS looks very fair to me, as the approach is tested against a variety of existing frameworks and evaluated in different categories. The results are reported in a concise way in the paper and in an extended way in the appendix. The evaluations seems to be carried out diligently and correctly. In terms of quality, AG–TS comes out on top (by a good margin). In terms of running time, it fares well. This shows that the approach is a very promising addition to or even improvement of existing frameworks. The only complaint I have is that the random experiments were just carried out three times, which seems like a rather low number to me.

The different parts of AG–TS are further tested in an ablation study, showcasing which parts contribute more to the performance than others. The full approach (which is used in the comparison against the other algorithms) has the best performance.

---

### Official Review · Reviewer_kk5B · 2023-05-08

**Potential Impact On The Field Of Automl Rating:** 3
**Technical Quality And Correctness Rating:** 3
**Clarity Rating:** 4

**Summary Of Contributions:**

The paper presents the part of the AutoGluon library dedicated for time series prediction. It targets the "A" part of the ABCD track and specifically highlights its ease of use. Its main working principles are ensembling of different models, ranging from statistical inference to neural networks, with an emphases on avoiding on-the-fly configuration during fitting. The paper features an experimental study of the library that includes several state-of-the-art time series prediction models, which it claims to outperform, and also an ablation study that tears off the components and sees what happens, which claims to show the necessity of most of the employed components.

**Actions Required To Increase Overall Recommendation:**

The most important thing to increase the overall recommendation is to address the two issues (statistical testing in the ablation study, and ~explanation of the preset derivation procedure~) that are detailed in "Technical Quality and Correctness".

**Clarity:**

All in all I enjoyed reading the paper quite a lot. Here is the list of minor issues I found.

Additional clarifications:

1) Please provide some guidelines for how past-only covariates can be used to improve predictions, as it is not immediately clear.

2) Page 8, "consider disabling the final ensembling step": please specify how you obtain the final predictions from multiple models in this case.

3) It would be good to add code examples that change the default preset, so that a user can get an idea for how difficult it is (or not) to configure the library in the case the default presets do not work well enough.

Typos and the like:

1) Page 2, second line of Section 2: the definition of $y_{i,1:T_i}$ features $T$ but not $T_i$, which is too early for Footnote 2 to work.

2) Page 3, "(The pandas development team, 2020)": this reference format looks really awkward with this kind of the author. This can also be said about the second such reference ("Joblib Development Team"), which is also given without any clarification (no URL etc). Maybe it is better to replace these entities with footnotes containing URLs.

3) Page 4: "by first first converting" => "by first converting".

4) Page 5: "The ensemble of the ensemble is...": something is definitely wrong here.

5) References: capitalization needs to be double-checked in things like "H2o automl", "m3", "m4", "m5", "Tpot", "xai", "Deepar" and many more.

**Overall Review:**

+ The paper presents a nice part of their library that is designed to perform state-of-the-art time series predictions using three (not overly long) lines of Python code.
+ The experimental comparison shows that it outperforms the competitors more often than not, while it is arguably easier to use.
+ The paper is written quite nicely, with only a little interventions remaining to improve the overall quality.
+ The ablation study is performed to evaluate the importance of the main components of the architecture of the proposed approach.
- The results of that ablation study were not evaluated for statistical significance, which is highly desired given how similar some ablated configurations are to the main one.
- ~The preset derivation procedure was not disclosed, which makes it complicated to judge whether these presets are not overfitted to the benchmarks employed in the comparison.~

**Potential Impact On The Field Of Automl:**

I need to note first that the contribution is a part of a library that appears to be known in the domain with some 50k downloads per month. So the penetration will definitely be higher compared to an isolated piece of software.

The claimed low complexity of just "three lines of Python code" to start using the library is quite attractive to the beginners indeed. The factual convenience of the use will depend, however, on how often the users will need to adjust the tunable knobs to get the result they want, which is difficult to estimate at the current stage.

I would estimate that the impact of this tool on the narrower field of AutoML will be rather medium, as it is more of a convenient wrapper of a carefully selected set of existing tools after all, but the effect on the normal end-users external to the field may be noticeably stronger.

**Review Confidence:**

3: You are fairly confident in your assessment. It is possible that you did not understand some parts of the submission or that you are unfamiliar with some pieces of related work.

**Review Rating:**

9: Strong Accept: Technically flawless submission with major impact, with no obvious flaws. Should be highlighted in the program.

**Review Summary:**

I find the current state of the paper to fit the description of a "technically sound paper with major impact, with perhaps some minor flaws". Due to its shape, it is likely to have a major impact on the end-users, and it will probably alter some default design decisions of the research in the AutoML community. ~The minor flaws are listed elsewhere in the review, and I believe that they are easily fixable, with a possibility to improve the final score to Strong Accept.~


**Technical Quality And Correctness:**

Generally, I see the paper nicely fitting the track, with the required points as follows:
1) It is a novel system that has features or application domains that were not available beforehand => first, it is a specialized time-series predictor within the library, and second, the particular set of models and the approach to fit them seems to be substantially different to the existing systems.
2) It already has an established user base => yes, definitely, and the enclosing library is already widely used across the society.
3) It is an open-source software package with an open-source software licence that allows users to easily use and contribute to it => yes, Apache 2.0.
4) It achieves excellent performance on the addressed application domains => yes, half of the paper is about that.

Given that, I still have identified a few issues to address that currently reduce the rating by one point. But they are quite easy to address, so this one is likely to increase.

1) ~Page 6, "Note that the presets were fixed a priori...": I think, the procedure to obtain the preset configuration (Table 6 in particular) should also be described, so that the readers can assess that e.g. none of the datasets used in performance evaluation have been used in it.~

2) Table 5: since the first three lines are very similar, especially the first two, I think some additional statistical test is necessary here to find out where the differences are significant.

---

### Official Review · Reviewer_MeQa · 2023-05-10

**Potential Impact On The Field Of Automl Rating:** 3
**Technical Quality And Correctness Rating:** 4
**Clarity Rating:** 4

**Summary Of Contributions:**

The paper presents AutoGluon-TimeSeries, a library for probabilistic time series forecasting that supplements Amazon Web Services' AutoGluon ML framework. The library includes several statistical and machine-learning models for time series forecasting as well as other established methods. An experimental study and comparison with competing, state-of-the-art frameworks demonstrates very good and often top rankings for AutoGluon, both in terms of quality and runtime.

**Actions Required To Increase Overall Recommendation:**

Elaborate on the impact of the software and describe its potential user base more clearly. Also, it would be interesting to discuss the role of purely quantitative time series forecasts, which is presently the main purpose of the software, and contrast it with qualitative predictions that might be achieved by modern ML techniques, including large language models.


**Clarity:**

The paper is very well written, easy to follow and includes all relevant information. There are only very minor issues with respect to punctuation.

**Overall Review:**

This is a significant paper that has the potential to advance the field of AutoML in the specific domain of probabilistic time series forecasting. The proposed library is based on best principles, comes with carefully chosen presets for the unexperienced user, and is thoroughly compared to the state of the art. Overall, the experimental data place it usually at the top of the field in terms of quality and runtime, except for some simple tasks that are solved faster with simple statistical models. Since my knowledge in the specific field is limited, I am a bit in doubt about the future impact of the contribution.

**Potential Impact On The Field Of Automl:**

There are already several libraries for time series forecasting on the market. However, the proposed library seems high performing and includes probabilistic time series, which are not covered by many existing approaches. Hence, I expect a moderate impact of the contribution.

**Review Confidence:**

3: You are fairly confident in your assessment. It is possible that you did not understand some parts of the submission or that you are unfamiliar with some pieces of related work.

**Review Rating:**

8: Accept: Technically sound paper with major impact, with perhaps some minor flaws.

**Review Summary:**

An essential flawless contribution adhering to all principles for an AutoML application paper. I expect the framework to have moderate to high impact on the field.

**Technical Quality And Correctness:**

The approach is based on modern design principles, is thoroughly evaluated and represents a high-quality application of ML. The experimental evaluation is fully reproducible, takes into account competing, state-of-the-art frameworks and uses a diverse selection of relevant benchmarks. The library is publicly available and follows established standards in documentation, testing, modularity and usability. The Apache 2.0 license allows free use and modifications of the software.

---

### Official Review · Reviewer_UcGF · 2023-05-10

**Potential Impact On The Field Of Automl:** 1. AutoGluon-TimeSeries has the poten…
**Potential Impact On The Field Of Automl Rating:** 4
**Technical Quality And Correctness Rating:** 4
**Clarity Rating:** 4

**Summary Of Contributions:**

1. The paper introduces AutoGluon-TimeSeries, an open-source AutoML library for probabilistic time series forecasting. It can generate both point and probabilistic forecasts for collections of univariate time series. AutoGluon-TimeSeries can also work with static and time-varying covariates, making it applicable to most real-world forecasting tasks.
2. The framework focuses on ease of use and robustness. It empowers users with limited expertise in the target domain to generate accurate point and quantile forecasts with just 3 lines of Python code. The architecture can handle failures of individual models when necessary, producing a valid result as long as any single model is trained successfully.
3. AutoGluon-TimeSeries combines conventional statistical models, machine-learning-based forecasting approaches, and ensembling techniques to achieve high accuracy within a short training time. It incorporates local models (e.g., ARIMA, Theta, and ETS) and global models (e.g., DeepAR, Temporal Fusion Transformer, LightGBM, and CatBoost) from both conventional statistical methods and deep learning-based approaches.
4. The paper presents design considerations, including ensembles over hyperparameter optimization (HPO), presets and default hyperparameters, and model selection for time series forecasting.
5. The paper evaluates the performance of AutoGluon-TimeSeries against other established forecasting methods and AutoML systems using 29 publicly available benchmark datasets. The results demonstrate the framework's strong performance, outperforming various competing approaches in terms of both point and probabilistic forecast accuracy.
6. The authors discuss future work, including exploring advanced ensembling strategies, improving calibration, supporting new problem types, and enhancing the scalability of the framework to handle larger-scale forecasting tasks.

**Actions Required To Increase Overall Recommendation:**

One Minor Correction:
There is a typo in the first paragraph on Page 5. The word "first" is repeated twice in the sixth line.

**Clarity:**

The paper is well-written and organized, with clear explanations of the proposed methods, the experimental setup, and the results.

One Minor Correction:
There is a typo in the first paragraph on Page 5. The word "first" is repeated twice in the sixth line.

**Overall Review:**

1. The paper presents a valuable contribution to the AutoML domain by introducing AutoGluon-TimeSeries, an open-source AutoML library for probabilistic time series forecasting.
2. It can generate both point and probabilistic forecasts for collections of univariate time series. AutoGluon-TimeSeries can also work with static and time-varying covariates, making it applicable to most real-world forecasting tasks.
3. AutoGluon-TimeSeries emphasizes simplicity and robustness. It is designed to be user-friendly, enabling users with limited domain expertise to generate accurate predictions with minimal coding effort.
4. AutoGluon-TimeSeries combines conventional statistical models, machine-learning-based forecasting approaches, and ensembling techniques to achieve high accuracy within a short training time.
5. The authors have demonstrated the effectiveness of AutoGluon-TimeSeries through extensive experiments on 29 benchmark datasets, showing that it outperforms other established methods and AutoML systems in terms of both point and probabilistic forecast accuracy. These results highlight the value of the proposed library as a powerful tool for practitioners and researchers in the field of AutoML.
6. The paper is well-written and organized, with clear explanations of the proposed methods, the experimental setup, and the results. This makes the paper accessible to a broad audience of researchers and practitioners.
7. The open-source nature of AutoGluon-TimeSeries can foster collaboration, inspire new ideas, and contribute to the growth of the AutoML community.

**Review Confidence:**

3: You are fairly confident in your assessment. It is possible that you did not understand some parts of the submission or that you are unfamiliar with some pieces of related work.

**Review Rating:**

9: Strong Accept: Technically flawless submission with major impact, with no obvious flaws. Should be highlighted in the program.

**Review Summary:**

The paper presents a comprehensive approach to probabilistic time series forecasting by introducing AutoGluon-TimeSeries, which leverages ensembles of diverse forecasting models to deliver high accuracy within a short training time. The paper demonstrates strong empirical results on 29 benchmark datasets, outperforming existing state-of-the-art approaches in terms of accuracy and computational efficiency. Furthermore, the paper is well-written and organized, making it accessible to a broad audience. The authors provide an open-source implementation of their method, which can promote further research and collaboration. Given the paper's positive aspects and potential for impact, I recommend its strong acceptance for publication.

**Technical Quality And Correctness:**

The technical quality and correctness of the presented AutoML library, AutoGluon-TimeSeries, is sound and of high quality. The authors have demonstrated the effectiveness of their approach through extensive experiments on 29 publicly available benchmark datasets. The results demonstrate the framework's strong performance, outperforming various existing state-of-the-art approaches in terms of both point and probabilistic forecast accuracy.

---

### Official Review · Reviewer_1hDb · 2023-05-11

**Potential Impact On The Field Of Automl Rating:** 4
**Technical Quality And Correctness Rating:** 3
**Clarity Rating:** 4
**Actions Required To Increase Overall Recommendation:** Adress some above-mentioned issues.

**Summary Of Contributions:**

The paper presents an open-source AutoML library for probabilistic time series forecasting.  Only with three lines of code, the library is able to generate accurate point and quantile forecasts.

**Details Of Ethical Concerns (Optional):**

/

**Clarity:**

The paper is well-written and structured. It is easy to follow and the discussions are supporting the experimental results.

**Overall Review:**

Positive: Nice presented methodology, with a lot of arguments and clarity on how it works, arguments for which methods are implemented, and enough details of explaining the results achieved.

Negative: In the future, please also test them on more publicly available repositories of datasets.

**Potential Impact On The Field Of Automl:**

The proposed library can bring a large impact on the AutoML community, especially on tasks related to time-series forecasting.

**Reproducibility (Optional):**

I have not checked it.

**Review Confidence:**

4: You are confident in your assessment, but not absolutely certain. It is unlikely, but not impossible, that you did not understand some parts of the submission or that you are unfamiliar with some pieces of related work.

**Review Rating:**

8: Accept: Technically sound paper with major impact, with perhaps some minor flaws.

**Review Summary:**

The paper is technically sound, explaining the concepts behind the pipeline and further providing arguments for all methods included in the modeling part. The paper is well-written and structured. It is easy to follow and the discussions are supporting the experimental results. The evaluation results are nicely presented.

**Technical Quality And Correctness:**

The paper is technically sound, explaining the concepts behind the pipeline and further providing arguments for all methods included in the modeling part. I like the concept of ensemble learning (the results are as expected).

Benchmarking with state-of-the-art tools has been performed on GulonTS datasets. Maybe in the future, you need to include more benchmark datasets available (maybe some from the UCR repository).